# Quasi-experimental controlled study protocol to reduce sedentary lifestyle in patients with type 2 diabetes

**Raquel Sainz-Prado**[1,2], **Andrea Sainz-Prado**[3], **Elena Andrade-Gómez**[1*],
**Beatriz Rodríguez-Roca**[4]

**1** Department of Nursing, University of La Rioja, Logroño, La Rioja, Spain, **2** Oncology Unit of Hospital San Pedro, La Rioja Health Service, Logroño, La Rioja, Spain, **3** Department of Neuroscience, University of the Basque Country, Leioa, Spain, **4** Department of Physiatry and Nursing, Faculty of Health Sciences, University of Zaragoza, Zaragoza, Spain

* elena.andrade@unirioja.es

## Abstract

### Introduction

The prevalence of diabetes has increased worldwide, making it the most prevalent metabolic disorder. Physical inactivity contributes to the progression of this disease and aggravates other comorbidities, such as obesity and cardiovascular disease. Beneficial strategies aimed at promotion and healthy aging, oriented to decrease sedentary behavior, are necessary to obtain desirable metabolic effects and improve the quality of life of people with diabetes.

### Objective

To examine, through a quasi-experimental study, the effect of decreasing sedentary time and increasing motivation to adopt an active lifestyle on different clinical, anthropometric and biochemical parameters in patients diagnosed with type 2 diabetes mellitus.

### Design and methodology

Quasi-experimental controlled study, single-center, parallel, two-branch, with a 12-month follow-up. We plan to recruit up to a total of 169 participants, who will be assigned in a 1:1 ratio to a control group (who will receive e-mails) and an intervention group (who will receive face-to-face group and individual visits and telephone calls). The active intervention has a duration of 6 months and will be carried out by nursing professionals. The primary outcomes are sedentary time measured with the Sedentary Behavior Questionnaire (SBQ) and the use of accelerometers, and the state of motivation to change measured through the Transtheoretical Model of Physical Exercise Change Questionnaire (TMPECQ).
**Trial registration.** ClinicalTrials.gov (NCT06893146).

**Data availability statement:** No datasets were generated or analysed during the current study. All relevant data from this study will be made available upon study completion.

**Funding:** The author(s) received no specific funding for this work.

**Competing interests:** The authors have declared that no competing interests exist.

## Introduction

Currently, more than 463 million people worldwide suffer from diabetes [1], with type 2 diabetes accounting for 90–95% of cases [2].

A state of recurrent hyperglycemia increases the Risk of chronic complications of diabetes, which include macrovascular (CVD, peripheral vascular disease, and lower limb amputations) and microvascular (retinopathy, nephropathy, and peripheral and autonomic neuropathy) complications [3,4]. In addition, prolonged habitual sedentary behavior also increases the risk of diabetic complications, even when considering physical activity levels [5]. This means that long periods of sedentary behavior also have negative effects, regardless of compliance with established physical exercise recommendations [6]. However, there is strong evidence that the association between sedentary lifestyles and all-cause mortality is more prevalent among physically inactive individuals [7].

Worldwide, approximately 52.3% of adults with type 2 diabetes exhibit prolonged sedentary behavior, with a higher prevalence in women (60.3%) than in men (56.5%) [8]. Additionally, recent observational studies have reported daily averages of 10 hours of sedentary time in this population [9,10]. Following this global evidence, a study conducted in Spain by the Spanish Diabetes Society (SED) and the Spanish Diabetes Federation (FEDE) identified that approximately four out of ten individuals with diabetes lead a sedentary lifestyle [11]. At the local level, nearly one-quarter of the population in La Rioja exhibits high levels of sedentary behavior, with women showing a higher prevalence. These findings highlight the importance of addressing this public health issue through targeted interventions [12].

Numerous research studies support the detrimental effects of these behaviors even in non-diabetics. In a study of sedentary individuals without diabetes who engaged in sedentary behavior for an average of 9 hours per day, it was observed that adding an additional 1 hour of sedentary behavior for 8 days was associated with a 22% increase in the likelihood of developing type 2 diabetes [13].

Recent studies in adults with type 2 diabetes show that interrupting prolonged periods of sitting with light activity, such as walking at low intensity for 3 minutes every half hour for 8 hours, reduces postprandial glucose, insulin, C-peptide, and triglyceride levels [14]. Replacing sitting time with standing (2.5 hours per day) and walking at low intensity (totaling 2 hours and 20 minutes per day) every 30 minutes improves 24-hour blood glucose levels and insulin sensitivity more than structures physical exercise [15]. Stair climbing has also been shown to be effective in lowering postprandial glucose levels [16,17], although it does not have a positive impact on glycosylated hemoglobin (HbA1c) [18]. A study of young people with obesity and glucose intolerance showed that 5-minute pause intervals every hour for 12 hours were more effective in reducing blood glucose levels than 1 hour of moderate-intensity exercise at the beginning of the day [19]. Another study indicated that short but intense exercise times (6 x 1-minute brisk walk at 90% of maximum heart rate) 30 minutes before main meals improved glycemic control in people with diabetes more than a single 30-minute session of moderate-intensity walking [20]. In general, any reduction in sitting time has been shown to be beneficial for people with type 2 diabetes [21].

This shows that, in addition to exercise, reducing sedentary lifestyle has a significant impact on the health and glycemic control of people of all ages with this condition. It is essential to promote programs that modify the lifestyle of the population, addressing the adoption and long-term maintenance of habits that reduce sedentary time. In addition to increasing physical activity, it is considered necessary to implement strategies to reduce sedentary behavior. This may be a more agile objective, serving as a complementary strategy to improve both physical fitness and metabolic health in individuals with type 2 diabetes.

In this regard, the novelty of the design lies in the structured integration of educational and motivational components with a specific focus on reducing sedentary behavior, developed and implemented from nursing practice, with both in-person and telephone follow-up. This approach enables personalized, sustainable, and adaptable care within real health-care settings, with the potential to improve adherence and quality of life in individuals with type 2 diabetes.

### Objectives

The general objective of the study is to know the effect of decreasing sedentary time and increasing motivation to adopt an active lifestyle of different health-related parameters in patients diagnosed with type 2 diabetes mellitus. Also to provide support and resources for the proper management of diabetes. Likewise, it will be verified that face-to-face interventions and telephone interventions by nursing professionals decrease the sedentary time in the diabetic population and if it means and improvement in clinical, anthropometric and biochemical parameters. Also, to evaluate the quality of life and the degree of motivation aimed at reducing sedentary time and to integrate the reduction of sedentary time in the lifestyle of the participants with practical tools and guidelines adapted to the circumstances.

## Methods and analysis

### Study design

A quasi-experimental controlled study with a 12-month follow-up will be carried out. The study has been chosen to be conducted as a single-center study for strategic and logistical reasons. A two-arm parallel design will be used. The control group (CG) will receive messages by mail with healthy lifestyle habits, while the intervention group (IG) will receive a behavioral intervention based on lifestyle modification, focusing on reducing sedentary time. These interventions will be directed by the research nurse.

### Sample and recruitment

The sample size was calculated for 230 members of the Diabetes Association of La Rioja (ASDIR) aged 18 years or older and diagnosed with type 2 diabetes mellitus.

A 95% confidence level and a 5% margin of error were used, values commonly accepted in community intervention studies [22]. Additionally, a dropout rate of 15% was anticipated, in line with that observed in previous studies conducted with populations with type 2 diabetes in similar contexts, where dropout rates range between 10% and 20% during multi-month interventions [23–26]. This estimation takes into account factors such as advanced age, disease burden, and the potential for low adherence to behavioral change programs. With these parameters, the final adjusted sample size is 169 individuals. Through a sequential allocation process based on the order of informed consent signing, approximately half will be assigned to the control group (CG) and the other half to the intervention group (IG), resulting in 84 participants in the CG and 85 in the IG.

To make it known to the associated and to know their willingness to participate, the president of the association will proceed to send emails and a broadcast message on WhatsApp through the community group. The dissemination of the project will begin in April 2025 and will last a total period of two months. Those interested and who meet the inclusion criteria will be selected during the following month for the study. A face-to-face visit will be made to the Association to explain

the detailed bases of the Project and to sign the informed consent, while the group allocation will be carried out simultaneously. The study will last for a total of 12 months from the date of informed consent signature. Fig 1 presents the flowchart outlining the planned process of sample selection, recruitment, and participant allocation for this study.

### Eligibility criteria

Patients will be eligible for inclusion if they have a diagnosis of type 2 diabetes mellitus, are 18 years of age or older and have minimal physical fitness. This is defined as the ability to sit down and stand up from a chair without assistance, walk short distances independently (at least 100 meters), and perform light household tasks—such as making the bed, washing dishes, or preparing simple meals—without difficulty. This information will be collected through clinical observation and self-report by the participant during the initial interview.

Patients with medical contraindications, pregnant women, individuals who have recently undergone major surgical interventions that limit functional mobility, or who, according to clinical judgment, have not achieved sufficient recovery, will be excluded. Patients undergoing cancer treatment or who have completed such treatment within the past year will also be excluded, as the side effects, in addition to the symptoms of the disease, may persist for months or even years [27,28]. Finally, individuals with cognitive impairment will be excluded, as cognitive symptoms can appear in early stages and affect the ability to participate in the intervention, [29–31], as well as those diagnosed with dementia or severe cognitive or psychiatric disorders.

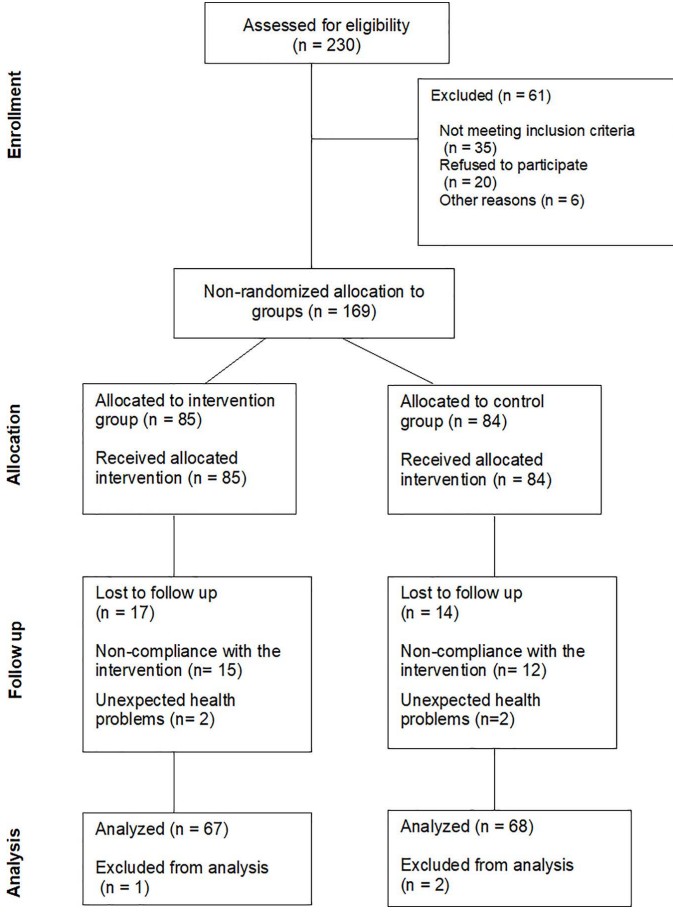

**Fig 1. Flow diagram of the inclusion process (TREND Statement).**

## Group allocation and masking

The distribution in the CG or IG will be performed sequentially in a 1:1 ratio in order of access to the study, i.e., the first patient who accepts the informed consent will be included in the IG, the next in the CG and so on until the sample is complete. This sequential assignment does not constitute strict randomization, and concealment techniques will not be employed. While this methodology is appropriate for research in real clinical settings, it may carry a risk of selection bias. This risk is acknowledged and will be considered when interpreting the results.

Due to the nature of the intervention, both the participants and the nursing staff and collaborating personnel will be aware of the assigned groups.

Additionally, the principal investigator will be responsible for data collection and analysis; therefore, outcome assessor blinding will not be applied.

## Study activities

The study has a total duration of 12 months. The intervention period is developed during the first 6 months. The flow diagram of the planned study procedure is detailed in Fig 2.

The CG will receive every two months detailed written information on healthy lifestyle habits by e-mail, emphasizing the importance of maintaining an active lifestyle, and will also be provided with standard educational material on the management of type 2 diabetes; this group will not receive additional interventions during the study period.

The IG will receive an intervention that will include two group sessions, bimonthly phone calls and a face-to-face visit with the study nurse.

The group sessions will be divided into two groups according to age, with a cutoff point at 65 years, and each subgroup will have the meeting on a different day, the first one will be held in the first month and the second one in the fifth month. This division will be implemented solely to facilitate the organization and convenience of the participants; both subgroups will receive the same content and duration in the sessions. The division does not imply any differences in the intervention but aims to optimize group dynamics; therefore, no differential effects are expected between the age subgroups.

The first session will focus on providing information about diabetes and sedentary behavior, explaining the health risks associated with sedentary habits as well as the benefits of adopting healthier lifestyles. Additionally, it will seek to empower participants to make informed decisions regarding their health.

The second session will be oriented towards reinforcing and consolidating the habits acquired, with an emphasis on identifying strategies for long-term maintenance, establishing sustainable routines, and preventing relapse. Each group meeting will have an approximate duration of 60–90 minutes.

Phone calls will be made every two months using motivational interviewing techniques with the aim of educating on habits and behaviors that reduce sedentary lifestyles, addressing needs and concerns on an individual basis, and enhancing patients' motivation, well-being and quality of life. The information will be self-reported and qualitative, aimed at supporting and adjusting the program according to the needs of each participant, rather than obtaining standardized quantitative measures.

The calls will last between 15 and 20 minutes. Participants will be informed that the sessions should conclude within this timeframe; however, calls may occasionally be extended if the 15-minute mark occurs at an inconvenient moment to end the conversation. Any topic related to diabetes management is open for discussion.

- First call: It will be ensured that the patient understands the program instructions, initial impressions and difficulties will be explored, and early adherence to sedentary behavior change will be assessed.

- Second call: Progress in reducing sedentary time will be evaluated, and emerging barriers or issues will be considered. Additionally, motivation will be reinforced.

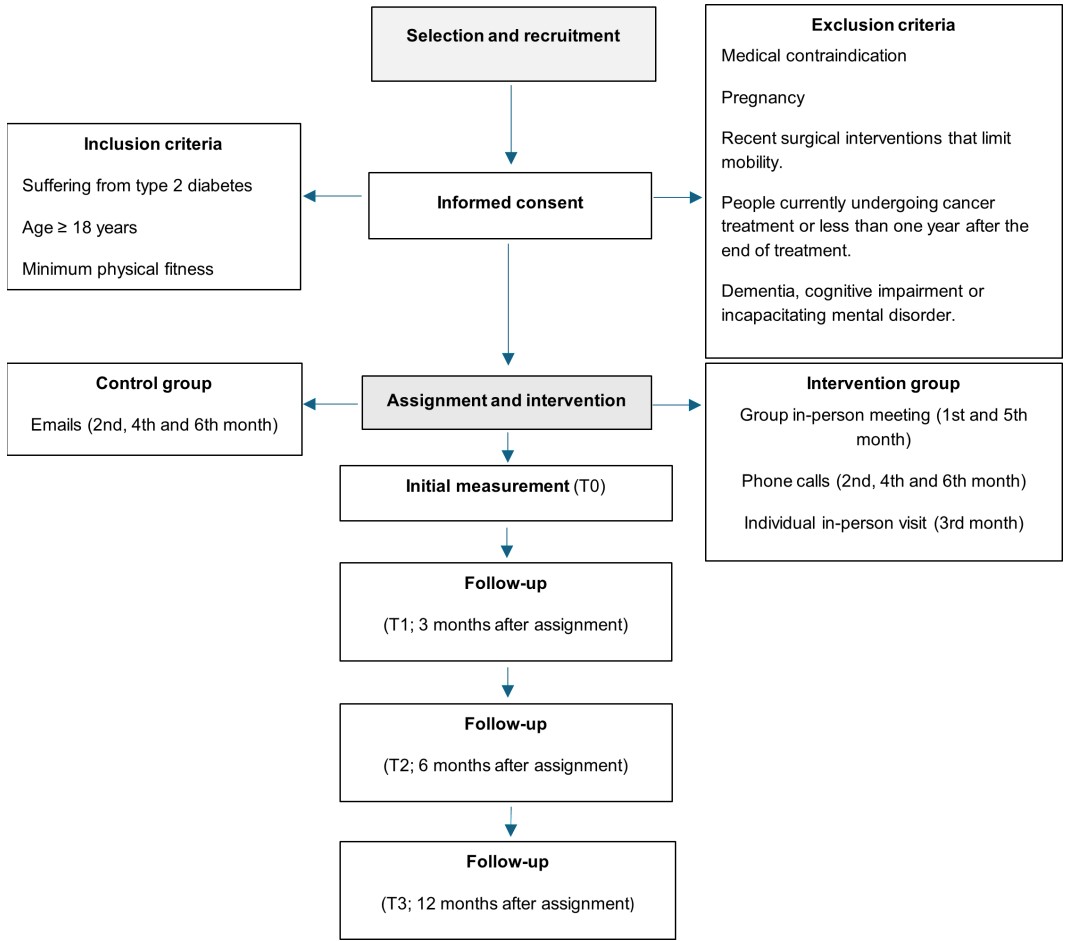

**Fig 2. Flow chart of the planned study procedure.**

- Third call: The overall experience and achievement of objectives will be reviewed.

If no response is obtained, up to three call attempts will be made, repeated over three consecutive days. If contact is not established after this period, the call will be considered missed, and the next call will be scheduled for two months later.

The motivational telephone interviews will be conducted by a nurse with complementary training in diabetes and motivational communication, together with a diabetes nurse educator who contributes clinical experience and specific training in the management of this disease. Both professionals have received specific training on the study protocol and intervention script, thereby ensuring the uniformity, quality, and fidelity of the calls made to participants.

The same purpose is served by the face-to-face visit that will be conducted in the third month of the study (Fig 3). During this individual session with the study nurse, progress in reducing sedentary behavior will be reviewed, specific questions will be addressed, and personalized emotional and motivational support will be provided, with the aim of enhancing the participant's perceived support in the self-care process. Additionally, the level of engagement with the program and the understanding of the proposed objectives may be directly assessed.

Measurements for all participants in the CG and the IG are taken at the beginning of the study (T0), at 3 months (T1), at an intermediate phase (6 months, T2) and at 12 months from the beginning of the study (T3). Once the 12 months have

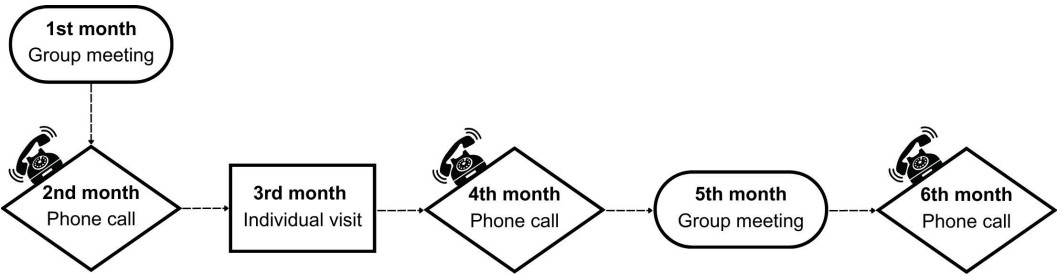

**Fig 3. Nursing interventions for the intervention group.**

elapsed, the study will be terminated. By then, data collection will have been completed, and the results are expected to be available within the following 3 months.

## Ethical approval

The present study is approved by the Ethics Committee for Drug Research in La Rioja (Ref. CEImLAR PI 778) and will comply with all established ethical standards. Healthcare professionals participating in the study will sign a document in which they undertake to protect the confidentiality of all patients. Before any procedure is performed, free and voluntary written informed consent must be obtained from each patient, which may be voluntarily revoked at any time during the study.

The information obtained in the study will be confidential and stored in coded form in an automated data file. The guidelines established in the Declaration of Helsinki, the current ethical guidelines for research involving human subjects and the Data Protection Act and the legislation in force in Spain will be followed. The processing of personal data of all patients will comply with the provisions of the Organic Law 3/2018 of December 5, 2018, on the Protection of Personal Data and Guarantee of Digital Rights and the provisions of Regulation (EU) 2016/679 of the European Parliament and of the Council of 27 April 2016 on Data Protection (GDPR). Our study design respects fundamental ethical principles and protects the welfare and rights of all participants.

Monitoring, audits, CEImLAR reviews and regulatory inspections will be allowed with the study, providing direct access to the original data.

Ethical audits are expected to be conducted annually and at the conclusion of the study. The research team will notify the ethics committee of any substantial protocol modifications through a written report sent via email, for evaluation and approval prior to implementation.

In the event of significant changes affecting participants, these will be communicated directly to them, and their informed consent will be requested again if necessary.

## Measurements and evaluations

Data will be collected at baseline (T0), three months (T1), interim (6 months = T2) and 12 months (T3) from the start of the study.

**Sociodemographic data.** Sociodemographic data will be collected on the participants, such as age, sex, place of residence, marital status, educational level, occupation, and employment status. They will be collected through the primary care clinical history and through a self-report that the participants will be asked to complete in the initial phase of the study (T0). These variables are documented in the S3 Appendix.

**Anthropometric data.** Anthropometric measurements of weight, height, waist-hip circumference are collected by healthcare personnel using calibrated instruments and clinical standard procedures. Weight is measured with shoes

and without heavy clothing. Height and weight are used to measure Body Mass Index (BMI) with the standardized formula, BMI = weight (kg)/ height (m)$^2$. Anthropometric measurements will be taken at each participant's health center of correspondence.

**Clinical characteristics.** Blood pressure and blood parameters are collected. In the latter, the levels of glucose, glycosylated hemoglobin (HbA1c), homeostasis model assessment-insulin residence (HOMA-IR), total cholesterol, high-density lipoprotein cholesterol (HDL-c), low-density lipoprotein cholesterol (LDL-c), triglycerides (TG), aspartate aminotransferase (AST), alanine aminotransferase (ALT) and gamma-glutamyl transferase (GAG) are examined, triglycerides (TG), aspartate aminotransferase (AST), alanine aminotransferase (ALT) and gamma-glutamyl transferase (GGT), creatinine, urea, creatinine clearance, iron, ferritin, sodium, potassium and C-reactive protein (CRP).

Blood samples will be drawn after a fasting period of at least 8 hours (no food or beverage intake) at the corresponding health center. Handling and transport will be conducted in accordance with standardized quality protocols, and the samples will be processed and analyzed in an accredited laboratory, following the current regulations of the Community of La Rioja. Biochemical analyses will be performed using automated techniques, employing equipment that is periodically calibrated and validated.

In relation to blood pressure, measurements will be taken at the corresponding health center. Readings will be obtained over the brachial artery, after 5 minutes of rest while seated and with the arms resting at heart level. Long-sleeved shirts will be removed for a correct reading and the cuff size must be adequate.

Data related to diabetes disease (year of diagnosis, antidiabetic treatment currently used, associated complications and frequency of Health care), smoking, alcohol consumption, past injuries and major surgeries, sedentary lifestyle, motivation to change physical exercise, physical activity and quality of life will also be collected through medical history and self-report.

**Level of sedentary lifestyle.** The level of sedentary lifestyle will be measured with the use of accelerometers and a questionnaire validated in Spanish. The questionnaire used is the Spanish version of the SBQ (S4 Appendix). This instrument measures the time spent in sedentary behaviors from the time the person gets up to the time he/she goes to bed, both during the week and at weekends. The responses are provided on numerical scales, and then the total number of hours spent on each activity is summed. The interclass correlation is 0.83–0.86 and the reliability index of the questions varies from 0.52 to 0.96 [32].

In addition, sedentary time will be evaluated quantitatively using accelerometers. The model used in our study is the ActiTrainer, which is small (8.56 x 3.81 x 1.52 cm) and light (48 g), ensuring great comfort during use. Previous studies have supported its use for an objective and more reliable measurement of the effectiveness of interventions [33–38] They will be placed by the study nurse during the measurement periods (T0-3), however, both the participants of the CG and the IG will receive training on their proper placement, removal and protocol. These should be worn at the waist, fastened with a well-fitting elastic belt, placed in the morning and removed at bedtime and during periods of exposure to water (bath, shower, swimming, etc.). To avoid skin irritation in case of incorrect fit, they may be worn over clothing. Participants must wear the device for 5 consecutive days, with a minimum of 10 waking hours per day.

This device only detects human movement, electronically filtering high-frequency vibrations produced by cars, buses, trains, etc. It has no other function than to measure physical activity through an encryption of internal operations that require specific software and cannot be used for any other purpose. It does not emit radiation or interfere with the human body. However, any eventuality must be reported.

Sedentary behavior will be defined as any period with ≤100 counts per minute (cpm) recorded by the ActiTrainer accelerometer. This threshold has been widely accepted in the scientific literature and specifically validated in adults against reference systems such as the IDEEA monitor, considered an almost gold standard for postural measurement, showing good agreement in estimated sedentary time [39]. Additionally, the cut-off points established by Freedson et al. (1998) will

be used to classify physical activity as light (101–1951 cpm), moderate (1952–5724 cpm), vigorous (5725–9498 cpm), and very vigorous (>9498 cpm), according to estimated energy expenditure in METs [40].

For the processing of the collected data, only days with a minimum of 10 hours of device wear time during waking hours will be considered valid. A minimum of 4 valid days, including at least one weekend day, will be required for the record to be deemed representative, in accordance with criteria established in previous studies [39,41]. Periods of non-wear, due to non-compliance or technical issues, will be excluded from the analysis based on validated algorithms for non-wear time detection, which automatically identify intervals of consecutive zero activity counts exceeding 60 minutes [42]. Should any participant fail to meet the minimum valid day criteria, a sensitivity analysis will be conducted, and the number of valid days per participant will be reported to ensure transparency and data quality.

**State of motivation to change.** Measured by the TMPEC questionnaire (S5 Appendix). This questionnaire aims to identify which stage of the transtheoretical model of change a person is in regarding his or her willingness to engage in physical exercise. The stages of change include precontemplation, contemplation, preparation, action, maintenance, and relapse. The questionnaire has items that use a respond scale from 1 (strongly disagree) to 5 (strongly agree) of reasons for not exercising, these items should only be completed by participants who do not exercise regularly. Similar items are rated in the table with the reasons why the participants perform physical exercise on a regular basis, the answers are rated on the same Likert-type scale from 1 to 5. The measurement of the questionnaire is performed by analyzing the average score of each corresponding stage of the transtheoretical model of change. Higher scores on each subscale of the questionnaire reflect a greater identification with the corresponding stage of behavioral change. Specifically, mean scores equal to or greater than 4 indicate a strong identification with that stage. A significant change can be interpreted as an increase of 0.5 points or more in the mean score of a subscale. Thus, increases in the mean scores of the preparation, action, and maintenance stages are considered indicative of progress in motivation toward the adoption and consolidation of regular physical exercise. Conversely, high scores on the precontemplation subscales reflect low readiness for change or resistance to modifying sedentary behavior. The internal consistency of this questionnaire shows Cronbach's Alpha values>0.70 for each of the factors resulting from the factor analysis: precontemplation (0.71), contemplation (0.71), preparation (0.75), action (0.71) and maintenance (0.70) [43].

**Physical activity level.** To measure the level of physical activity, the short, self-administered format of the "International Physical Activity Questionnaire" (IPAQ) (S6 Appendix) will be used. Questions are asked about the time spent in activity in the last 7 days, with a distinction between vigorous activity, walking and sitting. The results are calculated in terms of minutes per week, multiplying the number of days by the duration of the activity in minutes, obtaining a value in minutes per week for each of the categories. A person is considered inactive when he/she does not reach the recommended threshold of moderate physical activity (150 minutes per week), sufficiently active if he/she reaches 150 minutes of moderate activity or 75 minutes of vigorous activity per week and very active if he/she exceeds 300 minutes of moderate activity or 150 minutes of vigorous activity per week recommended by the WHO. The correlation coefficient of the short version of the questionnaire is 0.76 [44].

**Quality of life in people with diabetes.** The validated Spanish version of the Diabetes Quality of Life Questionnaire (DQoL) (S7 Appendix) will be used. It is a tool that measures quality of life in people with diabetes. Participants rate the items on a 5-point Likert-type scale, in which 1 reflects total disagreement and 5 reflects total agreement), other items are rated from 1 (very satisfied) to 5 (very dissatisfied). The results of the scores are quantified, a higher score indicates a better quality of life, while a lower score suggests a greater negative impact of diabetes on the patient's life. The internal consistency of this questionnaire shows Cronbach's alpha values of 0.68 to 0.86 [45].

## Statistical analysis

First, a descriptive analysis of the sociodemographic, clinical and anthropometric data of the participants will be performed. Quantitative variables (such as age, BMI, glucose, cholesterol and blood pressure levels) will be described using

means ± standard deviation (for those with normal distribution) and medians (with 25th and 75th percentiles) when they do not follow a normal distribution. Qualitative variables (such as sex, educational level, and marital status) will be described as relative frequencies, cumulative relative frequencies, and percentages. These analyses will be performed using Microsoft Excel for data organization and IBM SPSS Statistics for statistical calculations.

The normality of quantitative variables will be assessed using normality tests, such as the Shapiro-Wilk or Kolmogorov-Smirnov tests, as appropriate. To evaluate the homogeneity of variances and sphericity in repeated measures, Mauchly's test will be applied. To evaluate the efficacy of the intervention, the two groups of the study will be compared: CG and IG at different times (T0, T1, T2, T3). For continuous variables, a repeated measures analysis of variance (repeated measures ANOVA) will be conducted, applied only if the assumptions of residual normality and sphericity are met. In case of violation of the sphericity assumption, Greenhouse-Geisser or Huynh-Feldt corrections will be used to adjust the degrees of freedom. If the variables do not meet the normality assumption or the distribution is clearly non-parametric, the non-parametric Friedman test will be employed to analyze differences between repeated measurements.

The chi-square test will be applied to compare the distribution of qualitive variables between the two groups, such as the presence of comorbidities or the level of motivation to change. To establish statistical significance, a significance level of $p < 0.05$ will be used.

Regarding the handling of missing data, a descriptive analysis will be conducted to identify its pattern and proportion. An intention-to-treat (ITT) analysis will be adopted, including all participants assigned regardless of their adherence or losses during follow-up. For quantitative variables with missing data, multiple imputation will be applied using the Multiple Imputation by Chained Equations (MICE) method, generating several complete datasets that will be analyzed to improve accuracy and reduce bias. Additionally, a sensitivity analysis will be performed comparing results from imputed and complete data to assess the robustness of the conclusions.

One of the objectives of the study is to evaluate how the reduction of sedentary time is related to improvements in clinical parameters such as glucose, lipid and blood pressure levels. To this end, Pearson correlations will be used to analyze the relationship between sedentary time (measured with accelerometers) and changes in these parameters. In addition, multiple linear regression models will be used to adjust the analyses for possible covariates (age, sex, antidiabetic treatment, etc.).

Subgroup analyses based on relevant participant characteristics (such as age, sex, and level of motivation to change) will be considered exploratory and conducted with the aim of identifying possible differences in response to the intervention. To control for multiplicity across primary and secondary outcomes, the Bonferroni correction will be applied to adjust the significance level and minimize false positives. Given the relatively small number of comparisons, this conservative approach is appropriate and unlikely to substantially reduce statistical power. Results from these analyses will be interpreted with caution.

Changes in quality of life and motivation to change will be analyzed using the corresponding validated questionnaires. Both will be analyzed using repeated measures analysis of variance or linear regression to determine whether the interventions have a positive impact on these aspects over time.

The reliability and validity of the instruments used, such as the SBQ and the ActiTrainer accelerometer, have been verified. These instruments will measure sedentary lifestyle in an objective and reliable manner. On the other hand, the internal validity of the MTCEJ and DQOL Questionnaire has also been evaluated by psychometric analysis.

The results of the statistical analysis will be presented in tables and graphs to facilitate interpretation of the data, accompanied by confidence intervals to estimate the magnitude of the observed effects. Finally, the results will be discussed in relation to the existing literature on sedentary lifestyles in people with type 2 diabetes and the effectiveness of similar interventions.

## Discussion

This study will provide information of the efficacy of a nurse-led intervention focused on sedentary behavioral counselling to improve quality of life in people with type 2 diabetes mellitus.

The need for educational interventions in the health care setting stems from the alarming increase in the prevalence and mortality associated with chronic diseases, such as type 2 diabetes mellitus, and the increase in its risk factors. In this context, educational interventions have been consolidated as effective strategies to improve adherence to treatment, with telephone calls and periodic visits being some of the most evaluated and applied methods in various studies [46–48]. These interventions have not only been shown to be effective in improving adherence to treatment, but also in the management of chronic diseases, since they promote active and individualized control of patients' health [49].

It is important to note that education should not be understood as a one-time action, but as a continuous process that requires follow-up and personalized attention [50]. And to achieve successful management of patients with chronic diseases, such as diabetes, the focus should be on self-management, allowing patients to take an active role in their health care [51,52]. In this sense, our intervention combines face-to-face methods, both group and individual, with non-face-to-face strategies through telephone calls, as it constitutes a key strategy for personalized management. This modality makes it possible to address the specific problems and concerns of each patient during the process, generating a positive impact on their physical and psychological well-being, in addition to facilitating access to a greater number of people, regardless of socioeconomic level [53]. However, it is important to emphasize that, despite the benefits of non-face-to-face interventions, these should not be the only form of care. The lack of face-to-face interaction can make it difficult to build a trusting relationship between patient and health professional, which could decrease patient satisfaction and engagement with their treatment [54,55]. In addition, the use of digital technologies can generate insecurity in some patients, increasing levels of anxiety or depression, especially in those with little experience in handling new technological tools [56,57]. Therefore, it is crucial to complement remote interventions with face-to-face interventions, which allow a more comprehensive physical and emotional assessment, and strengthen the bond between nurse and patient [58].

The study's interventions have a personalized approach, which considers individual preferences and needs to consolidate the new habits acquired in the long term. Another strength of the study lies in the measurement of the main objective, as it is carried out using questionnaires and accelerometers, allowing a more objective and accurate measurement of the results. The study design, moreover, is oriented to be applicable in the health system, promoting a more efficient use of available resources. And the 12-month intervention, with an intensive focus in the first 6 months, can be easily managed by nurses. Furthermore, the study has a design that offers the possibility of generating solid, high-quality evidence on the benefits of the proposed intervention.

Nevertheless, this study presents certain limitations that must be considered. Firstly, the single-center design may restrict the generalizability of the results to other populations or healthcare settings. However, despite its single-center nature, the sample includes a broad and diverse community of individuals with type 2 diabetes from across the La Rioja region, providing sociodemographic variability and thereby strengthening the external validity of the findings within similar populations.

Secondly, the absence of formal randomization in participant allocation may introduce a certain selection bias. However, since assignment was conducted sequentially based on the order in which informed consent was signed, some nonsystematic variability in participant distribution may have occurred, potentially helping to reduce this risk. Even so, absolute comparability between groups at baseline cannot be guaranteed, which may partially affect internal validity. To minimize this bias, homogeneous inclusion criteria were defined, and a standardized follow-up protocol was applied to all participants. This design aspect should be considered when interpreting the results, as some differences may stem from uncontrolled baseline characteristics in addition to the effects of the intervention.

Thirdly, given that participants in both the control and intervention groups may share community or healthcare environments, there is a potential risk of contamination between groups, which could diminish the true difference observed in the effects. Additionally, the lack of blinding of both participants and researchers—a common limitation in behavioral studies—may affect the objectivity of the results and should be taken into account when interpreting the findings.

Moreover, the 12-month duration of the study entails a potential influence of seasonal variation on physical activity and sedentary behavior levels, an aspect that should be addressed in future research.

Finally, it is recommended to conduct a prospective follow-up extending beyond the 12-month period covered in this study, as assessing long-term outcomes would allow for understanding the sustainability of behavior changes related to sedentary lifestyles, as well as their impact on clinical parameters in individuals with type 2 diabetes. Such longitudinal evaluation is essential to determine the enduring effectiveness of behavioral interventions in this population.

From an economic perspective, the proposed intervention presents a potentially cost-effective profile for the healthcare system, as it focuses on education, promotion of healthy habits, and behavioral support—strategies that require a relatively low initial investment compared to other clinical or pharmacological interventions. Due to its ease of replication within nursing practice, the marginal costs associated with scaling the intervention are proportional and unlikely to compromise the financial sustainability of the system. While additional investment may be necessary for professional training, team coordination, and logistical adaptation, it is expected that the intervention will generate substantial medium- and long-term savings, primarily derived from reduced chronic complications, fewer hospitalizations, and improved disease management. This translates into a decreased care burden for health services, optimizing public resources and enabling their reallocation to higher-demand areas.

Moreover, the intervention contributes to improving care quality and promoting patient autonomy in disease management. Beyond generating new knowledge on community-based interventions aimed at reducing sedentary behavior in individuals with chronic diseases, this project pursues a clear practical application and transfer to the healthcare system of La Rioja, fostering innovation in nursing care, evidence-based decision making, and the sustainability of the system in the medium and long term.

## Supporting information

**S1 Appendix. TREND Checklist.**
(DOC)

**S2 Appendix. Trial study protocol approved by the ethics committee.**
(PDF)

**S3 Appendix. Data recording sheet.**
(DOCX)

**S4 Appendix. Sedentary Behavior Questionnaire.**
(DOCX)

**S5 Appendix. Transtheoretical Model of Physical Exercise Change Questionnaire.**
(DOCX)

**S6 Appendix. International Physical Activity Questionnaire.**
(DOCX)

**S7 Appendix. Quality of Life Questionnaire in Type 2 Diabetes.**
(DOCX)

## Acknowledgments

We thank the University of Zaragoza for their collaboration in providing the accelerometers that will be used in the study.

## Author contributions

**Conceptualization:** Raquel Sainz-Prado, Andrea Sainz-Prado, Andrade Gómez Elena, Beatriz Rodríguez-Roca.

**Methodology:** Raquel Sainz-Prado, Beatriz Rodríguez-Roca.

**Supervision:** Andrade Gómez Elena, Beatriz Rodríguez-Roca.

**Visualization:** Raquel Sainz-Prado.

**Writing – original draft:** Raquel Sainz-Prado, Andrade Gómez Elena, Beatriz Rodríguez-Roca.

**Writing – review & editing:** Raquel Sainz-Prado, Andrea Sainz-Prado, Andrade Gómez Elena, Beatriz Rodríguez-Roca.

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
