## [Decision Letter · Decision Letter 0]

12 May 2025

PONE-D-25-06236Clinical trial protocol to reduce sedentary lifestyle in patients with type 2 diabetesPLOS ONE

Dear Dr. Elena,

Thank you for submitting your manuscript to PLOS ONE. After careful consideration, we feel that it has merit but does not fully meet PLOS ONE’s publication criteria as it currently stands. Therefore, we invite you to submit a revised version of the manuscript that addresses the points raised during the review process.

We look forward to receiving your revised manuscript.

Kind regards,

Hidetaka Hamasaki

Academic Editor

PLOS ONE

Journal Requirements:

*2. * When completing the data availability statement of the submission form, you indicated that you will make your data available on acceptance. We strongly recommend all authors decide on a data sharing plan before acceptance, as the process can be lengthy and hold up publication timelines. Please note that, though access restrictions are acceptable now, your entire data will need to be made freely accessible if your manuscript is accepted for publication. This policy applies to all data except where public deposition would breach compliance with the protocol approved by your research ethics board. If you are unable to adhere to our open data policy, please kindly revise your statement to explain your reasoning and we will seek the editor's input on an exemption. Please be assured that, once you have provided your new statement, the assessment of your exemption will not hold up the peer review process.

Reviewers' comments:

Reviewer's Responses to Questions

**Comments to the Author**

1. Does the manuscript provide a valid rationale for the proposed study, with clearly identified and justified research questions?

Reviewer #1: Yes

Reviewer #2: Yes

2. Is the protocol technically sound and planned in a manner that will lead to a meaningful outcome and allow testing the stated hypotheses?

Reviewer #1: Yes

Reviewer #2: Yes

3. Is the methodology feasible and described in sufficient detail to allow the work to be replicable?

Reviewer #1: Yes

Reviewer #2: Yes

4. Have the authors described where all data underlying the findings will be made available when the study is complete?

Reviewer #1: Yes

Reviewer #2: Yes

5. Is the manuscript presented in an intelligible fashion and written in standard English?

Reviewer #1: Yes

Reviewer #2: Yes

6. Review Comments to the Author

You may also provide optional suggestions and comments to authors that they might find helpful in planning their study.

Reviewer #1: The authors plan to conduct a quasi-experimental study with a target of 169 participants for six months of active intervention to examine the effect of sedentary time and motivation to adopt an active lifestyle on various health-related parameters in T2D patients.

1. Line 107. “half will be allocated to the control group…” Please clarify how participants will be assigned to different groups.

2. Line 145. The IG group will be divided into two groups based on age. Please be more specific. Also how will the age cutpoint be selected? Also, will there be age/group-specific effect or heterogeneity?

3. Please comment on the generalizability of a single-center study as proposed in this study.

Reviewer #2: This manuscript presents a clinical trial protocol addressing a significant public health issue—reducing sedentary lifestyles among patients with type 2 diabetes mellitus. The proposed intervention, leveraging nurse-led motivational and educational strategies, holds substantial promise. Strengths include a clear design, detailed methodological procedures, and the use of both subjective and objective measures of sedentary behavior. However, several critical areas need clarification or further methodological rigor to enhance transparency and ensure replicability. Here are some comments to consider:

Introduction

1. Clearly specify the global and local prevalence of sedentary behavior in type 2 diabetes patients (lines 23-28) to highlight the relevance of the study in the chosen setting.

2. Explicitly mention the novelty of your intervention: e.g., the combined use of face-to-face group sessions, individual consultations, and regular telephone follow-ups specifically delivered by nurses.

Methods

Study design and sample calculation (lines 89-130)

1. Clarify the rationale behind choosing a 5% margin of error and anticipated 15% dropout rate; consider referencing similar studies or standard guidelines.

2. Provide more details regarding randomization procedures; clarify if allocation concealment techniques (e.g., sealed envelopes, computer-generated sequences) are used to avoid selection bias (lines 131-136).

Eligibility criteria (lines 125-130)

1. Define clearly what constitutes “minimal physical fitness” as an inclusion criterion; specify measurable criteria or validated instruments.

2. Consider explaining why certain exclusion criteria (recent surgery, oncological treatment, cognitive impairment) were selected and how these might impact study outcomes.

Interventions and activities (lines 137-175)

1. Specify the frequency, duration, and detailed content of face-to-face group sessions and individual visits to enhance reproducibility.

2. Clarify who will conduct motivational interviewing by phone, and provide their training background or qualifications to ensure intervention consistency.

Outcome measurements (lines 180-280)

1. Sedentary behavior measurement:

• Clearly define the thresholds for classifying sedentary versus active behaviors from accelerometer data.

• Specify how missing accelerometer data (e.g., due to non-compliance or technical issues) will be handled.

2. Motivation to change:

• Briefly clarify the clinical interpretation of scores from the TMPEC questionnaire—what specific score ranges indicate meaningful shifts in motivational stages.

3. Clinical parameters:

• Specify if blood parameters will be collected under fasting conditions and detail the handling and analysis methods (e.g., laboratory accreditation, biochemical assay types).

Statistical Analysis (lines 280-329)

1. Clearly justify the selection of repeated measures ANOVA or Friedman’s test; specify conditions under which each test will be applied (normality checks, variance homogeneity tests).

2. Explain your approach to handling missing data clearly; state whether techniques like intention-to-treat analysis, multiple imputation, or sensitivity analyses will be used.

3. Clarify if subgroup analyses are pre-specified or exploratory, and specify how multiple testing (e.g., Bonferroni or Benjamini-Hochberg corrections) will be handled to minimize type I errors.

Ethical Considerations (lines 156-179)

1. Confirm the planned frequency of ethical audits and how protocol amendments will be communicated to the ethics committee and participants.

Discussion (lines 330-380)

1. Explicitly acknowledge potential limitations, including generalizability due to single-center design, the risk of contamination between groups, and the possible impact of seasonal variation on sedentary behavior.

2. Recommend prospective follow-up beyond the 12-month period to evaluate the long-term sustainability of behavior changes and clinical outcomes.

3. Discuss possible economic implications or cost-effectiveness analysis for implementing such nurse-led interventions at scale within health systems.

Overall, addressing these points will enhance the scientific rigor, reproducibility, and transparency of this promising clinical trial protocol.

7. PLOS authors have the option to publish the peer review history of their article (what does this mean? ). If published, this will include your full peer review and any attached files.

**Do you want your identity to be public for this peer review?** For information about this choice, including consent withdrawal, please see our Privacy Policy .

Reviewer #1: No

Reviewer #2: **Yes: ** Sepehr Khosravi

---

## [Author Response · Author response to Decision Letter 1]

23 Jun 2025

Raquel Sainz, PhD

Editor-in-Chief

PLOS One

Logroño, June 21, 2025

Dear Editor-in-Chief,

We sincerely thank the Editorial Board for their consideration in our manuscript and the reviewers for their insightful and constructive feedback. We have addressed all the reviewers' suggestions to improve the manuscript and have implemented revisions throughout.

We are truly grateful for the feedback received, which has significantly contributed to refining the clarity, methodological rigor, and overall quality of the manuscript.

We are pleased to submit this improved version of the manuscript for your consideration for publication in PLOS ONE. Alongside the revised manuscript, we are including a document that highlights all modifications, as well as a detailed response to each of the reviewers' comments explaining how we addressed their feedback as follows.

Sincerely,

On behalf of all the authors,

Raquel Sainz

University of La Rioja

Logroño, La Rioja

Response to Referee #1

We sincerely appreciate the time devoted to reviewing our manuscript and the valuable comments provided, which have helped us to clarify important aspects of the study. We have carefully addressed each of the points raised and made the corresponding changes to the text, which are described below.

(1) “Half will be allocated to the control group…” Please clarify how participants will be assigned to different groups.

We have clarified in the manuscript that the assignment of participants to the groups will be carried out sequentially and alternately, based on the order in which the informed consent is signed. That is, the first participant to sign the consent form will be assigned to the intervention group, the second to the control group, and so on. In addition, we have specified the number of participants in each group: 85 in the intervention group and 84 in the control group. (Lines 126-129)

(2) The IG group will be divided into two groups based on age. Please be more specific. Also how will the age cutpoint be selected? Also, will there be age/group-specific effect or heterogeneity?

The cut-off point at 65 years of age was established solely for organizational and logistical reasons, as this age is a commonly used threshold to distinguish between the adult and older adult populations in the healthcare field. This division facilitates the formation of more generationally homogeneous groups and allows for better management and comfort during group sessions, without implying any differences in the intervention’s content or focus. (Lines 182-188)

(3) Please comment on the generalizability of a single-center study as proposed in this study.

We appreciate your comment regarding the generalizability of the study. While we acknowledge that this design may entail certain limitations concerning external validity, we consider that in our case this limitation is mitigated by the fact that the sample includes a large proportion of the type 2 diabetes community in the La Rioja region, through the ASDIR. This extensive coverage within a single regional community provides considerable sociodemographic diversity, thereby enhancing the representativeness of the sample and strengthening the applicability of the findings to similar populations. Moreover, the components of the intervention (in-person health education and telephone follow-up) are easily reproducible and do not require complex resources, which facilitates their implementation in other similar settings. (Lines 491-496)

Response to Referee #2

We sincerely appreciate your kind words regarding the manuscript and the recognition of the proposed intervention, as well as your acknowledgment of the potential impact this protocol may have on public health improvement for patients with type 2 diabetes. We deeply value your critical and constructive observations, which have been instrumental in enhancing the quality and clarity of the manuscript. We have carefully reviewed each of the points raised and made the necessary revisions to strengthen the transparency and reproducibility of the study. Below, we detail how we have addressed each of the specific comments.

SPECIFIC POINTS

Introduction

(1) Clearly specify the global and local prevalence of sedentary behavior in type 2 diabetes patients to highlight the relevance of the study in the chosen setting.

The following information has been added in lines 56–61: In Spain, a study conducted by the Spanish Diabetes Society (SED) and the Spanish Diabetes Federation (FEDE) identified that approximately four out of ten people with diabetes lead a sedentary lifestyle (1).

At the local level, nearly one-quarter of the population in La Rioja exhibits high levels of sedentary behavior, with women showing the highest prevalence (2).

Although specific local data disaggregated by chronic diseases are not available, these figures highlight the need to develop local studies that allow for the design of interventions tailored to the characteristics and needs of this population.

(2) Explicitly mention the novelty of your intervention: e.g., the combined use of face-to-face group sessions, individual consultations, and regular telephone follow-ups specifically delivered by nurses.

We have added an explanation in the introduction highlighting the novelty of the intervention (lines 91–96). This novelty lies in the combination of educational and motivational components, along with both telephone and in-person follow-up, all conducted by nursing staff. This comprehensive design, based on educational and motivational tools, facilitates personalized, sustainable behavior change that is potentially applicable to real-world healthcare settings.

Methods

(1) Clarify the rationale behind choosing a 5% margin of error and anticipated 15% dropout rate; consider referencing similar studies or standard guidelines.

A 95% confidence level and a 5% margin of error were used—parameters widely accepted in epidemiological and community intervention studies as the standard to ensure precise estimation with a reasonable probability of error (3). This choice allows for a balance between statistical accuracy and the practical feasibility of the study.

Furthermore, an anticipated attrition rate of 15% was considered based on similar studies involving adult populations with chronic conditions such as type 2 diabetes, where dropout rates range from 10% to 20% during interventions lasting several months (4–7). This estimate takes into account factors such as advanced age, disease burden, and potential low adherence to behavior change programs. (Lines 119-125)

(2) Provide more details regarding randomization procedures; clarify if allocation concealment techniques (e.g., sealed envelopes, computer-generated sequences) are used to avoid selection bias.

We have clarified in the manuscript that participant allocation will be conducted using a systematic alternate method (1:1) according to the order of inclusion, without concealment techniques, and therefore it is not a strict randomization. Additionally, it has been noted that the principal investigator will be responsible for both data collection and analysis, so assessor blinding will not be applied. These limitations have been taken into account in the study design and will be addressed in the discussion of the results. (Lines 164-170)

Eligibility criteria

(1) Define clearly what constitutes “minimal physical fitness” as an inclusion criterion; specify measurable criteria or validated instruments.

Thank you for your valuable observation. In response to your comment, we have clarified in the manuscript the definition of the concept of "minimum physical fitness" as an inclusion criterion. Specifically, it is defined as the functional ability to rise from and sit down in a chair unassisted, walk short distances (at least 100 meters) independently, and perform light household tasks without difficulty (e.g., making the bed, washing dishes, or preparing simple meals). This fitness is assessed through clinical observation and participant self-report during the initial interview, which ensures that participants have the minimum physical capacity required to safely take part in the intervention. (Lines 144-149)

(2) Consider explaining why certain exclusion criteria (recent surgery, oncological treatment, cognitive impairment) were selected and how these might impact study outcomes.

We have included a more precise explanation regarding these exclusion criteria in lines 150-158 of the manuscript.

Recent surgery

Regarding the criterion of recent surgical interventions that limit mobility, it has been refined to improve its accuracy and applicability, focusing on recent major surgical procedures that result in functional limitations or that, based on clinical judgment, have not allowed sufficient recovery to participate in the study.

Functional recovery after surgery largely depends on the type of procedure, the patient’s baseline condition, and the presence of comorbidities. Furthermore, the restoration of strength, mobility, and participation in daily or recreational activities may be compromised for weeks or even months depending on the complexity of the surgery and potential postoperative complications.

Since the intervention proposed in this study requires a minimum level of mobility for proper implementation and to ensure participant safety, patients who have undergone recent major surgeries are excluded. Major surgery is defined as procedures requiring general or regional anesthesia that may compromise functional mobility for a prolonged period (e.g., hip or knee arthroplasty, spinal surgery, major abdominal or thoracic surgery, among others). Patients with minor or rapidly recovering surgeries (such as uncomplicated appendectomy or laparoscopic cholecystectomy) are not unnecessarily excluded, provided they have regained their usual functional mobility and present no limitations that interfere with participation in the intervention. This measure also aims to ensure sample homogeneity and the validity of the results obtained.

Patients undergoing active oncological treatment

Chemotherapy is a common component of treatment for many oncological diseases at some stage of their course. Unfortunately, antineoplastic drugs can cause side effects that negatively impact patients’ quality of life. These effects may occur immediately (hours or days), early (days or weeks), delayed (weeks or months), or late (months or years). Among them, asthenia — a persistent feeling of physical, emotional, and mental exhaustion — is one of the most prevalent complications, affecting up to 90% of patients during treatment and persisting in more than 50% after its completion (8). Other frequent symptoms include pain, psychological disturbances, sleep disorders, gastrointestinal issues, anorexia, and toxicity (9). These symptoms, resulting from both the disease and its treatment, could limit patients’ active participation in the intervention and compromise the validity of the outcomes obtained.

Cognitive impairment

Cognitive impairment may represent a prodromal stage of Alzheimer’s disease, especially when manifested as mild cognitive impairment. Although significant impact on activities of daily living may not be observed in many cases, evidence shows that this condition is associated with difficulties in acquiring verbal and non-verbal information, language skill impairments, attention problems, executive dysfunction, and reduced cognitive flexibility — cognitive symptoms that can appear from early stages (10–12).

For this reason, cognitive impairment has been considered an exclusion criterion, as it may hinder understanding and adherence to the intervention and could interfere with the reliability of the responses provided in the questionnaires used.

Interventions and activities

(1) Specify the frequency, duration, and detailed content of face-to-face group sessions and individual visits to enhance reproducibility.

The requested details have been incorporated into the manuscript and can be found in lines 189-215 and 222-228. These sections precisely describe the frequency, content, and approximate duration of the in-person group sessions and individual visits. Additionally, detailed information regarding the purpose of the telephone calls has been included.

(2) Clarify who will conduct motivational interviewing by phone, and provide their training background or qualifications to ensure intervention consistency.

The requested information has been added to the manuscript (see lines 216-221). The motivational telephone interviews will be conducted by a nurse with a master’s degree in biomedical research and additional training in diabetes and motivational communication, together with a diabetes nurse educator with extensive experience and specialized training in this disease. Both professionals have been trained in the protocol and interview script to ensure the consistency and quality of the intervention.

Outcome measurements

(1) Sedentary behavior measurement:

1.1. Clearly define the thresholds for classifying sedentary versus active behaviors from accelerometer data.

The following information has been added to the manuscript, in lines 324-332. Sedentary behavior will be defined as any period with ≤100 counts per minute (cpm) recorded by the ActiTrainer accelerometer. This threshold has been widely accepted in the scientific literature and specifically validated in adults against reference systems such as the IDEEA monitor, considered an almost gold standard for postural measurement, showing good agreement in estimated sedentary time (13). Additionally, the cut-off points established by Freedson et al. (1998) will be used to classify light physical activity (101–1951 cpm), moderate (1952–5724 cpm), vigorous (5725–9498 cpm), and very vigorous (>9498 cpm), according to the estimated energy expenditure in METs (14).

1.2. Specify how missing accelerometer data (e.g., due to non-compliance or technical issues) will be handled.

Thank you for your observation. For data processing, only days with a minimum of 10 hours of device wear time during waking hours will be considered valid. At least 4 valid days (including at least one weekend day) will be required for the recording to be deemed representative, following criteria established in previous studies (13–15). Periods of non-wear, due to non-compliance or technical issues, will be excluded from the analysis using validated algorithms for non-wear detection, which automatically identify periods without use based on consecutive intervals of no movement recordings longer than 60 minutes (16). If any participant does not meet the minimum valid days criterion, a sensitivity analysis will be conducted and the number of valid days per participant will be reported to ensure transparency and data quality. This information has been incorporated into the manuscript, in lines 333-342.

(2) Motivation to change:

Briefly clarify the clinical interpretation of scores from the TMPEC questionnaire—what specific score ranges indicate meaningful shifts in motivational stages.

As stated in the manuscript, the interpretation of the TMPEC questionnaire results is carried out by comparing the mean scores obtained in each subscale corresponding to a stage of change. Although the questionnaire is validated in Spanish (17), there is no exact numerical threshold established to define significant changes. However, from a clinical perspective, an increase of 0.5 points or more in the mean score of a subscale may be considered indicative of a significant change in motivation. Additionally, a shift in the highest score from one stage to another (for example, from “Contemplation” to “Preparation”) suggests progression in the motivational process. Mean scores above 4 reflect a strong identification with that stage. (Lines 354–362)

(3) Clinical parameters:

Specify if blood parameters will be collected under fasting conditions and detail the handling and analysis methods (e.g., laboratory accreditation, biochemical assay types).

Blood parameters will always be collected under fasting conditions, after a minimum period of 8 hours without intake of food or beverages. The samples will be processed and analyzed in a laboratory

---

## [Decision Letter · Decision Letter 1]

4 Jul 2025

PONE-D-25-06236R1Clinical trial protocol to reduce sedentary lifestyle in patients with type 2 diabetesPLOS ONE

Dear Dr. Elena,

Thank you for submitting your manuscript to PLOS ONE. After careful consideration, we feel that it has merit but does not fully meet PLOS ONE’s publication criteria as it currently stands. Therefore, we invite you to submit a revised version of the manuscript that addresses the points raised during the review process.

We look forward to receiving your revised manuscript.

Kind regards,

Hidetaka Hamasaki

Academic Editor

PLOS ONE

Journal Requirements:

Reviewers' comments:

Reviewer's Responses to Questions

**Comments to the Author**

1. Does the manuscript provide a valid rationale for the proposed study, with clearly identified and justified research questions?

Reviewer #1: Yes

Reviewer #2: Yes

2. Is the protocol technically sound and planned in a manner that will lead to a meaningful outcome and allow testing the stated hypotheses?

Reviewer #1: Yes

Reviewer #2: Yes

3. Is the methodology feasible and described in sufficient detail to allow the work to be replicable?

Reviewer #1: Yes

Reviewer #2: Yes

4. Have the authors described where all data underlying the findings will be made available when the study is complete?

Reviewer #1: Yes

Reviewer #2: Yes

5. Is the manuscript presented in an intelligible fashion and written in standard English?

Reviewer #1: Yes

Reviewer #2: Yes

6. Review Comments to the Author

You may also provide optional suggestions and comments to authors that they might find helpful in planning their study.

Reviewer #1: Thanks for your response addressing all the raised concerns. I do not have further questions or comments.

Reviewer #2: Most of my earlier points have been addressed; however, two items still need further revision, and I have one new comment:

1 Introduction

You added national figures (Spain 40 %, La Rioja ≈ 25 %), but a global estimate is still missing. Please insert a concise line with a reliable worldwide figure for sedentary behaviour in adults with T2DM—for example, “≈ 60 % of adults with type 2 diabetes sit ≥ 8 h/day” (cite an authoritative source).

2 Methods – Study design / sample size

Allocation is now sequential and non-concealed, which is appropriate only for a quasi-experimental design. Yet the manuscript alternates between “quasi-experimental” and “clinical trial.”

• Option A: adopt true randomisation (computer-generated sequence + sealed opaque envelopes), or

• Option B: keep sequential allocation, label the study consistently as a “quasi-experimental controlled study,” and discuss the resulting selection-bias risk in Methods and Limitations.

3 Statistical analysis – multiplicity (new)

Please state how multiplicity will be controlled for the full set of primary and secondary outcomes (e.g., Bonferroni, Benjamini-Hochberg) or justify why no adjustment is necessary.

7. PLOS authors have the option to publish the peer review history of their article (what does this mean? ). If published, this will include your full peer review and any attached files.

**Do you want your identity to be public for this peer review?** For information about this choice, including consent withdrawal, please see our Privacy Policy .

Reviewer #1: No

Reviewer #2: **Yes: ** Sepehr Khosravi

---

## [Author Response · Author response to Decision Letter 2]

16 Jul 2025

Dear Editor-in-Chief,

We appreciate the opportunity to revise our manuscript and would like to sincerely thank the Editorial Board for their continued consideration. We also wish to thank the reviewer for the valuable comments that have helped us further refine our work.

In response to the suggestions provided, we have made the corresponding modifications to improve the manuscript. In the following pages, we detail our point-by-point responses, explaining the changes made and how each comment has been addressed.

We are pleased to resubmit the revised version of our manuscript for your consideration for publication in PLOS ONE, with the hope that the improvements introduced meet the expectations of the journal.

Thank you for your time and kind consideration.

Sincerely,

On behalf of all the authors,

Raquel Sainz

University of La Rioja

Logroño, La Rioja

Response to Referee #2

We are grateful for your thorough review and thoughtful suggestions. We have carefully considered each of your comments and implemented the necessary revisions, as detailed below.

(1) You added national figures (Spain 40 %, La Rioja ≈ 25 %), but a global estimate is still missing. Please insert a concise line with a reliable worldwide figure for sedentary behaviour in adults with T2DM—for example, “≈ 60 % of adults with type 2 diabetes sit ≥ 8 h/day” (cite an authoritative source).

Updated data regarding sedentary behavior in adults with type 2 diabetes mellitus worldwide have been incorporated into the manuscript (Lines 57-60). Specifically, the results of a systematic review and meta-analysis published in 2024, which included 12 studies conducted in various countries with a total sample of 1,446,076 individuals, estimated that the prevalence of prolonged sedentary behavior among patients with type 2 diabetes is 52.3%, with higher rates in women than in men (60.3% vs. 56.5%) (1). The Maastricht Study reported that patients with type 2 diabetes spend an average of 10.1 hours per day engaged in sedentary behaviors (2). These findings are consistent with other recent studies, which also indicate high levels of sedentary behavior, with daily averages close to 10 hours (3).

(2) Allocation is now sequential and non-concealed, which is appropriate only for a quasi-experimental design. Yet the manuscript alternates between “quasi-experimental” and “clinical trial.”

• Option A: adopt true randomisation (computer-generated sequence + sealed opaque envelopes), or

• Option B: keep sequential allocation, label the study consistently as a “quasi-experimental controlled study,” and discuss the resulting selection-bias risk in Methods and Limitations.

On Friday, March 7, we received a message from the editorial staff stating the following:

“The in-house editorial staff feels that your study meets the World Health Organization definition of a clinical trial because it is a prospective study in which participants were assigned to 'Nurse directed behavioral intervention' to investigate the effects on 'Anthropometric measurements/ Blood pressure and blood parameters'. Please change your manuscript’s article type to ‘Clinical Trial’ when you resubmit your manuscript.”

Additionally, we were requested to submit the CONSORT statement despite already having uploaded the TREND checklist to the submission platform. This caused some confusion, as the TREND format is more appropriate for quasi-experimental studies.

However, we would like to clarify that the study has consistently followed a non-randomized design, with sequential and non-concealed allocation. Therefore, we have decided to maintain this allocation method and, accordingly, to consistently label the study as a “quasi-experimental controlled study” throughout the manuscript.

Information regarding this aspect has been incorporated into lines 166–170 of the Methods section. Furthermore, a discussion of the potential risk of selection bias inherent to this design has been included in lines 501–510 of the Limitations section. We acknowledge that the lack of full randomization may impact internal validity; therefore, we have detailed the strategies implemented to minimize this bias and contextualized how this affects the interpretation of the results.

In line with this correction, we have updated the study title, removed the CONSORT checklist, and uploaded the TREND checklist, which better suits the methodological characteristics of our study.

Additionally, since the study is registered at ClinicalTrials.gov, the information has also been updated in the submission platform, and the version of the manuscript includes the registration reference.

We sincerely apologize for any confusion this may have caused and thank you for your understanding and guidance.

(3) Please state how multiplicity will be controlled for the full set of primary and secondary outcomes (e.g., Bonferroni, Benjamini-Hochberg) or justify why no adjustment is necessary.

We appreciate the comment regarding the control of error due to multiple testing. After careful consideration, we have chosen to apply the Bonferroni correction to adjust the significance level in the variable analyses, in order to minimize the risk of false positives and ensure that the results are reliable and robust. Given that the number of comparisons performed is relatively small, we consider this conservative correction to be feasible without excessively compromising statistical power. Furthermore, we believe that this strategy is appropriate in our clinical context, where the accuracy and reliability of the results are essential to guide future recommendations for the intervention of patients with type 2 diabetes (Lines 433-438).

REFERENCES

1. Salari N, Ahmadi M, Ghasemi H, Yarani R, Mohammadi M. The global prevalence of sedentary time in diabetes and metabolic syndrome: a systematic review and meta-analysis. Iran J Public Health. 2024 Sep;53(9):2020–9. https://doi.org/10.18502/ijph.v53i9.16455

2. Van der Berg JD, Stehouwer CD, Bosma H, Van der Velde JH, Willems PJ, Savelberg HH, et al. Associations of total amount and patterns of sedentary behaviour with type 2 diabetes and the metabolic syndrome: The Maastricht Study. Diabetologia. 2016 Apr;59(4):709-18. https://doi.org/10.1007/s00125-015-3861-8

3. Zhang X, Yang D, Luo J, Wang Y, Zhang Y, Li X, et al. Determinants of sedentary behavior in community-dwelling older adults with type 2 diabetes based on the behavioral change wheel: a path analysis. BMC Geriatr. 2024;24:502. https://doi.org/10.1186/s12877-024-05076-0

---

## [Decision Letter · Decision Letter 2]

1 Aug 2025

Quasi-experimental controlled study protocol to reduce sedentary lifestyle in patients with type 2 diabetes

PONE-D-25-06236R2

Dear Dr. Elena,

We’re pleased to inform you that your manuscript has been judged scientifically suitable for publication and will be formally accepted for publication once it meets all outstanding technical requirements.

Kind regards,

Hidetaka Hamasaki

Academic Editor

PLOS ONE

Additional Editor Comments (optional):

Reviewers' comments:

Reviewer's Responses to Questions

**Comments to the Author**

1. If the authors have adequately addressed your comments raised in a previous round of review and you feel that this manuscript is now acceptable for publication, you may indicate that here to bypass the “Comments to the Author” section, enter your conflict of interest statement in the “Confidential to Editor” section, and submit your "Accept" recommendation.

Reviewer #1: (No Response)

2. Is the manuscript technically sound, and do the data support the conclusions?

Reviewer #1: (No Response)

3. Has the statistical analysis been performed appropriately and rigorously? 

Reviewer #1: (No Response)

4. Have the authors made all data underlying the findings in their manuscript fully available?

Reviewer #1: (No Response)

5. Is the manuscript presented in an intelligible fashion and written in standard English?

Reviewer #1: (No Response)

6. Review Comments to the Author

Reviewer #1: (No Response)

7. PLOS authors have the option to publish the peer review history of their article (what does this mean? ). If published, this will include your full peer review and any attached files.

**Do you want your identity to be public for this peer review?** For information about this choice, including consent withdrawal, please see our Privacy Policy .

Reviewer #1: No

---

## [Editor Report · Acceptance letter]

PONE-D-25-06236R2

PLOS ONE

Dear Dr. Elena,

I'm pleased to inform you that your manuscript has been deemed suitable for publication in PLOS ONE. Congratulations! Your manuscript is now being handed over to our production team.

Kind regards,

on behalf of

Dr. Hidetaka Hamasaki

Academic Editor

PLOS ONE